# Atherosclerosis, Cardiovascular Disorders and COVID-19: Comorbid Pathogenesis

**DOI:** 10.3390/diagnostics13030478

**Published:** 2023-01-28

**Authors:** Yulia A. Makarova, Varvara A. Ryabkova, Vladimir V. Salukhov, Boris V. Sagun, Aleksandr E. Korovin, Leonid P. Churilov

**Affiliations:** 1Laboratory of the Microangiopathic Mechanisms of Atherogenesis, Saint Petersburg State University, 199034 Saint-Petersburg, Russia; 2M.V. Chernorutsky Department of Internal Medicine (Hospital Course), Pavlov First Saint Petersburg State Medical University, 197022 Saint-Petersburg, Russia; 3N.S. Molchanov 1st Clinic for the Improvement of Physicians, S.M. Kirov Military Medical Academy, 194044 Saint-Petersburg, Russia; 4Department of Experimental Tuberculosis, Saint Petersburg Research Institute of Phthisiopulmonology, 191036 Saint-Petersburg, Russia

**Keywords:** atherosclerosis, COVID-19, inflammation, cardiovascular system, cytokines, endothelium, lipoproteins, renin-angiotensin system, atheroma, autoimmunity, vasa vasorum

## Abstract

The article describes how atherosclerosis and coronavirus disease 19 (COVID-19) may affect each other. The features of this comorbid pathogenesis at various levels (vascular, cellular and molecular) are considered. A bidirectional influence of these conditions is described: the presence of cardiovascular diseases affects different individuals’ susceptibility to viral infection. In turn, severe acute respiratory syndrome coronavirus 2 (SARS-CoV-2) can have a negative effect on the endothelium and cardiomyocytes, causing blood clotting, secretion of pro-inflammatory cytokines, and thus exacerbating the development of atherosclerosis. In addition to the established entry into cells via angiotensin-converting enzyme 2 (ACE2), other mechanisms of SARS-CoV-2 entry are currently under investigation, for example, through CD147. Pathogenesis of comorbidity can be determined by the influence of the virus on various links which are meaningful for atherogenesis: generation of oxidized forms of low-density lipoproteins (LDL), launch of a cytokine storm, damage to the endothelial glycocalyx, and mitochondrial injury. The transformation of a stable plaque into an unstable one plays an important role in the pathogenesis of atherosclerosis complications and can be triggered by COVID-19. The impact of SARS-CoV-2 on large vessels such as the aorta is more complex than previously thought considering its impact on vasa vasorum. Current information on the mutual influence of the medicines used in the treatment of atherosclerosis and acute COVID-19 is briefly summarized.

## 1. Introduction

Soon after the outbreak of COVID-19 in Wuhan, China, it became clear that patients with cardiovascular diseases (CVD) had a higher risk of acute complications [1]. The cardiovascular system is one of the main targets of the SARS-CoV-2 virus, resulting in the increased incidence of severe disorders including myocarditis, pericarditis, arrhythmias, heart failure and thromboembolism in COVID-19 cases. The mortality rate of such patients ranges from 11% to 19% [1]. Mortality from COVID-19 among people with CVD is much higher than average. In the presence of CVD accompanied by hypertroponinemia (for example, against the background of severe coronary heart disease), it exceeds 70% in some samples. Up to 25% of COVID-19 cases are accompanied with the development of cardiovascular complications, mainly among older people with pre-existing atherosclerosis and its clinical manifestations. CVDs are most prevalent comorbidities in patients with COVID-19 and most strongly affect mortality [2]. In total, heart failure accounts for approximately 40% of all deaths in patients with COVID-19. It should be noted that mortality in the presence of CVD is higher in elderly and senile patients, which is due a greater prevalence and severity of atherosclerosis and its manifestations in these age groups [3].

In the very beginning of the COVID-19 pandemic, autopsy studies described fulminant myocarditis with features of direct viral and immunopathologically mediated heart damage [4]. It was assumed that myocarditis is common in COVID-19, but further studies on larger samples with magnetic resonance imaging (MRI) reported a more modest prevalence of severe myocarditis (accompanied by systolic dysfunction, electrocardiogram (ECG) changes and an increase in myocardial cell injury biomarkers in the blood) [5]. Thus, out of more than 168,000 patients hospitalized with COVID-19 in Florida, myocarditis was diagnosed only in 0.4% of cases [6]. Apparently, vascular lesions in COVID-19 are more significant than the cytotoxicity of the virus in cardiomyocytes. Of course, cardiovascular implications of COVID-19 are most dangerous for those who, on the basis of preexisting atherosclerosis, already have chronic lesions of the coronary/cerebral arteries and marginally reduced perfusion reserves of the myocardium and other vital organs [1].

Atherosclerosis is a chronic disease of the elastic and muscular-elastic arteries, characterized by the deposition of atherogenic lipoproteins in the vascular wall, phagocytic and proliferative, a synthetic reaction to these deposits from the cells of the vascular wall and mononuclear cells migrating there from the bloodstream and the resulting self-sustaining inflammation [7]. Therefore, a pro-inflammatory and thrombophilic state is an integral feature of atherosclerosis, potentially increasing vulnerability to severe COVID-19 because the underlying endothelial dysfunction might represent the ideal deregulated immunological setting in which SARS-CoV-2 triggers a ‘‘cytokine storm’’. Even despite thromboprophylaxis, heart attacks, strokes and venous thromboembolic complications in the Milan cohort of people hospitalized with COVID-19 in 2020, for example, developed in more than 8% of cases [8].

Regarding long-term effects of COVID-19, more than 76% of survivors who had severe or moderate disease still had signs of cardiovascular dysfunction detected with in-depth cardiac imaging 2 months after recovery [8]. Some of these individuals had not had a diagnosis of any cardiovascular disease before the pandemic. At the beginning of the pandemic, Vinciguerra et al. suggested a mutual relationship between SARS-CoV-2 and atherosclerosis [9]. Atherosclerosis is provoked by damage to the endothelium, leading to disruption of the homeostasis of the vascular wall. However, the subsequent exacerbation of the existing endothelial dysfunction may affect different individual susceptibility to viral infection [9].

On the other hand, the systemic hyperinflammatory response in COVID-19 leads to shifts in the ratio of hemostasis and antihemostasis and vasculities, which potentially favor atherogenesis and accelerate the formation of atherosclerotic plaque [7], as well as raises the risk of its complications. Increased production of pro-inflammatory mediators with their excessive systemic action can lead to the activation of proteases, causing, in particular, degradation of the fibrous capsule of atherosclerotic plaques [9]. Studies [1,9,10] have shown that infection with SARS-CoV-2 can accelerate the course of both initial and advanced stages of atherogenesis and cause its clinical manifestation by different mechanisms. These mechanisms include activation of coagulation and thrombosis, secretion of pro-inflammatory cytokines and chemokines by endothelial cells, rise in the level of fibrinogen, antithrombin and D-dimer in the blood and, ultimately, triggering regional disseminated intravascular coagulation [10].

Today it is clear that the interaction of COVID-19 and cardiovascular pathology is multifactorial, but it is not always obvious why in some patients the infection is accompanied by severe cardiovascular manifestations or leaves a significant cardiovascular mark as part of the post-COVID syndrome, while in others this does not happen.

In this review, we discuss possible mechanisms of cardiovascular injury in COVID-19, listed in Table 1.

## 2. Direct and Immune-Related Cytopathic Effects of SARS-CoV-2 on Cardiomyocytes

The SARS-CoV-2 S-protein can directly bind to human ACE2 to enter cells. SARS-CoV-2 has a polybasic insertion (PRRAR) at the S1/S2 cleavage site that can be cleaved by furin. This new cleavage site, absent in SARS-CoV-1 or other coronaviruses, appears to facilitate processing of the S protein at the S1/S2 boundary which is required for SARS-CoV-2 entry into cells. Similar cleavage sites have been described for highly pathogenic avian influenza and Newcastle disease viruses. There is a hypothesis that this remarkable feature plays a significant role in the multicellular tropism of SARS-CoV-2, contributing to the multiorgan effects of COVID-19 [19].

In vitro infection of cardiomyocytes with the S1 subunits of the SARS-CoV-2 spike protein can alter their transcriptome, induce fragmentation of myofibrils and destruction of nuclei. Thus, the S1 protein itself is dangerous for cardiac cells, and SARS-CoV-2 is characterized by the ability to cause hypertrophic myocardial remodeling, cardiac dysfunction and myocarditis [20]. In coronavirus myocarditis (as in myocarditis caused by enteroviruses and herpes group viruses), both lymphocytic infiltration of the myocardium and autoantibodies against cardiac antigens are detected [21]. This testifies in favor of the autoimmune component of delayed myocardial damage in COVID-19, probably with the contribution of the molecular mimicry phenomenon and an immunostimulatory adjuvant-like effect of hypercytokinemia [22]. The pathogenesis of myocarditis in COVID-19 seems to include not only immune-mediated inflammatory cytotoxicity, but also dysfunction of receptors and ion channels caused by autoantibodies. A number of functional blocking and/or stimulating autoantibodies to various proteins, including those expressed in the heart, have been identified in patients with severe COVID-19. [23] Autoimmune phenomena in COVID-19, including those related to the cardiovascular system, can be facilitated by a shift in the differentiation of T-helpers towards Th1 cells, which occurs against the background of hypercytokinemia [24] (see below).

It should be also noted that myocarditis cases after receiving mRNA-based COVID-19 vaccines (with largely favorable outcomes) have been reported worldwide, especially in adolescents and young adults [25]. Potential mechanisms of myocarditis in this case include the activation of pro-inflammatory cascades in the heart by viral mRNA, molecular mimicry between the spike protein of SARS-CoV-2 and cardiac self-antigens and sex hormone-related factors.

Finally, the death of cardiomyocytes can be caused by excessive exposure to certain cytokines, both coming from inflammatory cells that have infiltrated the heart and those circulating in the blood at extremely high concentrations under cytokine storm conditions ([16] and see below). A multiple increase in serum levels of IL-2, IL-6, IL-10, GCSF, IFN-γ, MCP-1, MIP-1-α and TNF-α seems to contribute to myocardial injury in progressive hemodynamic shock of any etiology, for example, in chimeric antigen receptor (CAR) T cell therapy-induced cytokine release syndrome [26]. It is known that myocardial damage and left ventricular systolic dysfunction are often detected after the immunostimulatory use of CAR-T, and the sooner from the onset of hypercytokinemia the IL-6 inhibitor tocilizumab was administrated, the less adverse cardiovascular effects were recorded [26].

## 3. Respiratory Failure and Mixed Hypoxia

The primary target of coronaviruses is the respiratory system. Seasonal low-pathogenic coronaviruses infect the upper respiratory tract, while highly pathogenic ones also affect its lower parts [27,28]. These highly pathogenic coronaviruses, including SARS-CoV-2, induce the destruction of type II pneumocytes, causing the respiratory component of hypoxia. Another potential mechanism for hypoxia was noted by Liu et al. [29]. SARS-CoV-2 has ORF8, ORF10, orf1ab, ORF3a proteins and surface glycoproteins that interact with the heme group of hemoglobin. ORF8 proteins can bind to porphyrin in the heme. The proteins ORF10, orf1ab and ORF3a can remove iron from the heme molecule in the β1 chain of hemoglobin. Thus, hemoglobin loses its ability to efficiently deliver oxygen, and, in addition to respiratory hypoxia, hemic hypoxia occurs. Inactivation of hemoglobin leads to the deformation of erythrocytes and worsening of their rheological characteristics; that serves as the basis for impaired microcirculation and the stagnant component of hypoxia [29]. It is worth recalling that any prolonged hypoxia always tends to involve a histotoxic mechanism of its pathogenesis, because the swelling of mitochondria during hypoxic necrobiosis of any etiology deprives them of full functionality [30]. This is especially true for COVID-19 against the background of CVD, due to the presence of mitochondrial dysfunction in these diseases. The role of mitochondrial dysfunction in the comorbidity of COVID-19 and CVDs may be very significant. There have been reports of the involvement of mitochondria in the activation and regulation of the innate immune response, as well as in the intensification of inflammation and transition to its chronic form (including in COVID-19) [31]. However, the mitochondrial phospholipid cardiolipin is a component of atheromas and a target of an autoimmune response in a number of diseases with impaired hemostasis. For example, autoantibodies to cardiolipin are present in atherosclerosis and aneurysms of the abdominal aorta [32]. Damage to mitochondria with the release of their DNA (mtDNA) is a potential result of SARS-CoV-2 infection. Damaged mitochondria are sources of products of incomplete oxygen reduction, i.e., reactive oxygen species (ROS), which play an important role in the pathogenesis of atherogenesis [33].

SARS-CoV-2 infection may also promote telomere shortening by exporting the TERF2IP-TRF2 complex following mitochondrial ROS (mtROS)-mediated activation of the ribosomal s6-kinase (p90RSK). Thus, it is possible that oxidative stress caused by SARS-CoV-2 infection may contribute to aging phenotypes that exacerbate the development and progression of atherosclerosis in COVID-19 survivors [20]. This is consistent with the theory of atherogenesis as a manifestation of local clonal aging and deterioration of the vascular cells [7].

So, mixed hypoxia in COVID-19 includes all of its known endogenous pathogenetic types, from respiratory to histotoxic, and that is why it can be torpid to non-complex treatment and can contribute to the manifestation of latent heart failure in people with atherosclerosis who survived COVID-19.

## 4. Endothelial Dysfunction in COVID-19 and Atherosclerosis: Common Links

Arterial endothelium is the site of key dolipid changes during atherogenesis [7]. However, dysfunction of endotheliocytes and even vascular cell death also occurs early in COVID-19, since SARS-CoV-2 is endotheliotropic [8,12,34]. Damage to the endothelium can be caused both by a direct cytopathic effect of the virus, and by the effects of parietal leukocytes involved in the immune response to the virus. Several pathways are involved in the development of endothelial-mediated complications of COVID-19 [8,34,35].

Under physiological conditions, the endothelium retains anticoagulant, antithrombotic and fibrinolytic properties. When stimulated by inflammatory and infectious triggers, the balance can shift in the opposite direction due to the expression of tissue procoagulant and thrombogenic factors, the release of von Willebrand factor (VWF) from Weibel–Palade endothelial bodies, the production of thromboxanes and plasminogen activator inhibitor-1 (PAI-1) [36]. These and other factors may be involved in the recruitment of leukocytes from the blood, causing a prolonged pro-inflammatory and thrombophilic state [12]. Inflammatory mediators released by marginated blood cells and the endothelium and activated in plasma (not only cytokines, but also components of the complement, kinin, coagulation, and fibrinolytic systems and eicosanoids) are the key factors that contribute to the disruption of endothelial function [37]. Thus, proteomic analysis of 185 biomarkers of inflammation and endothelial dysfunction in the systemic circulation showed that the presence of a cytokine storm in COVID-19 was combined with diffuse damage to the vascular endothelium [38]. An increase in the content of pro-inflammatory cytokines in the systemic circulation in patients with COVID-19 directly correlates with an increase in serum concentrations of markers that reflect the development of systemic vasculitis and vascular bed remodeling processes [39]. In addition, there is a relationship between the clinical severity of COVID-19 on the one hand and the risk of development and severity of endothelial dysfunction on the other [40]. In COVID-19, there is a procoagulant shift, which manifests itself in the increase in serum levels of fibrinogen, fibrin breakdown products, D-dimer and von Willebrand factor, which correlates with the severity of the disease and the risk of thrombosis [41]. When activated during COVID-19 by pro-inflammatory cytokines, endothelial and blood cells generate free oxygen and halogen-containing radicals, contributing to the development of oxidative stress. However, the latter is also a recognized factor in the progression of atherosclerosis [7,36]. The fact is that ROS, as well as halogen-containing radicals of leukocytes, modify low-density lipoproteins (LDL) into their oxidized form (oxLDL). Moreover, such LDL serve as neoantigens and trigger autoimmune processes in atheromas. In addition, they activate Toll-like receptors on vascular wall cells, which leads to the self-assembly of inflammasomes and the generation of new pro-inflammatory cytokines from inactive precursors. Finally, modified LDL are characterized by interaction with scavenger receptors of macrophages and unregulated uptake by these cells, which increases the cholesterol load on macrophages without adequate compensation mechanisms that are triggered only when LDL are absorbed through a specialized apoB receptor [7]. Chronic inflammation in the intima can increase the rate of LDL deposition in the subendothelial layer of blood vessels, thereby enhancing atherogenesis [7,33]. OxLDL, as already noted, serve as ligands for the Toll-like type 4 receptor (TLR4), which is expressed by various cell types in the walls of vessels affected by atherosclerosis. Activated TLR4 on macrophages can trigger signaling cascades that induce the release not only of proinflammatory cytokines, but also of proteases [42]. TLR4 and TLR2, in response to components of dying cells (alarmins), as well as to the SARS-CoV-2 virus, can initiate the expression of MyD88, a key protein in pro-inflammatory signaling. Analysis of differentially expressed genes in cardiomyocytes has shown that this protein is the most important “coupler” of atherogenesis and pathogenesis of COVID-19. Protein MyD88, either by activating the transcription factor NF-κB or by itself, initiates the expression of pro-inflammatory cytokines and chemokines, CXCL1, CXCL2, CXCL3, CXCL8 and IL-1β, which promotes local inflammation in the myocardium and in the vascular wall, as well as excessive systemic hypercytokinemia, up to its extreme degree, “cytokine storm”, sometimes resulting in the hemodynamic shock [1,34].

For more than 110 years, since the biochemical studies of atheromas by A. Windaus (1910) and the first rabbit model of atherosclerosis created by S.S. Khalatov and N.N. Anichkov (1912), atherosclerosis has been associated with cholesterol and the response of arterial wall cells to its chronic excess [7]. However, cholesterol, as a modulator of the fluidity of cell membranes, can directly affect the penetration of viruses into cells.

Many viruses use cholesterol-rich regions of the host cell’s membrane (the so-called “lipid rafts”) for internalization and self-assembly. In vitro studies have shown that the number and location of ACE2 receptors used by SARS-CoV-2 as gates for intracellular entry correlated with the content of cholesterol and such “rafts” in the cell membrane. Viruses have cholesterol receptor proteins. Scavenger class B receptor type 1 (SR-B1) can provide the S1 subunit of the spike protein of coronaviruses with binding to high-density lipoproteins and promote the spread and entry of SARS-CoV-2 into cells along with these metabolites. Intracellular cholesterol in macrophages of CVD patients activates the NLRP3 inflammasome, which contributes to the development of both atherosclerosis and cytokine storm in COVID-19 [43].

Another pathogenetic mechanism for endothelial dysfunction in COVID-19 may be SARS-CoV-2-mediated damage to the vascular endothelial glycocalyx (VEGLX)—glycosylated lipid-protein molecules that coat the vascular endothelium and play an important role in vascular homeostasis (in particular, maintaining the physiological negative charge of the endothelium) [36].

## 5. The Role of Dysfunction of the Renin–Angiotensin–Aldosterone System

It is known that SARS-CoV-2 uses ACE2, a negative regulator of the renin–angiotensin–aldosterone system (RAAS), as a receptor for entry into cells, including almost all cell types in the heart and blood vessel wall. ACE-2 is a type I transmembrane protein that is expressed in the lungs (high level of expression on the surface of type II alveolar cells). The physiological role of ACE-2 is primarily associated with the breakdown of angiotensin I (Ang-II) to the inactive Ang-(1-9) peptide, which is further converted into Ang-(1-7), using ACE or other peptidases, which binds to Mas receptors. Ang-(1-7) provides vaso- and cardioprotection, antiproliferative, anti-inflammatory and natriuretic effects, and also has protective effects against heart failure, thrombosis, myocardial hypertrophy, fibrosis, arrhythmia and atherogenesis [13,44].

In COVID-19 with a high viral load, the expression of this protein decreases. As a result, the activity of Ang-(1-7), which is a product of the ACE2-dependent proteolytic pathway, decreases. A decrease in the level of Ang-(1-7) and an increase in the concentration of angiotensin II (Ang-II) implies dysregulation of the RAAS with hypernatriemia, hypokalemia, increased vasoconstriction, acceleration of tissue fibrosis, smooth muscle cells proliferation and pro-inflammatory events in the vessel wall. Most of these events are mediated by the hormones of the mineralocorticoid zone of the adrenal cortex (primarily, aldosterone) [2,17,18,44,45]. Effects of the increased aldosterone activity contribute to the development of both heart failure and arterial hypertension, as well as to the acceleration of atherogenesis, with an increased risk of its complications [2,17,18,44,45]. Moreover, hypokalemia leads to tachyarrhythmias [7]. Ang-II is closely associated with atherosclerosis, e.g., it induces the formation of abdominal aortic aneurysms (atypical complications of atherosclerosis) in mice with dyslipidaemias [34]. Thus, RAAS dysfunction is a common link in the pathogenesis of COVID-19, atherosclerosis and their complications.

Thus, thrombogenic events reported in COVID-19—whether thromboembolism and disseminated intravascular coagulation or atherosclerotic plaque progression—serve as promoters of both COVID-19 and atherosclerosis as well as their complications.

## 6. Miscellaneous Mechanisms SARS-CoV-2-Dependent Vasculopathy

It was found that SARS-CoV-2 effectively infects cells of the immune system even with a low level of ACE2 expression, such as macrophages (including macrophages in the vascular wall) and T-lymphocytes (which infiltrate the vascular wall in atherogenesis). SARS-CoV-2 can potentially bind to the transmembrane glycoprotein CD147, which belongs to the immunoglobulin superfamily and can provide an additional route of infection [46].

It has been recently hypothesized that the dipeptidyl peptidase-4 (DPP-4) receptor facilitates SARS-CoV-2 entry into cells as it has similar spike glycoprotein to Middle East respiratory syndrome (MERS-CoV), which invades human cells using this receptor. The same DPP-4 is also known for its role in atherogenesis, since it influences migration of the macrophage lineage cells, in particular, monocytes, and suppresses the production of the antiatherogenic lipokine adiponectin [12].

Brain-derived neurotrophic factor (BDNF) serves as a proatherogenic mediator and is overexpressed in atheromas [47]. The same cytokine is a predictive marker of severe COVID-19 [48], but at the same time negatively correlates with severe post-COVID cognitive impairment [49].

## 7. Excessive Systemic Action of Pro-Inflammatory Autacoids

Severe COVID-19 is often associated with acute circulatory failure with blood flow centralization. However, in terms of critical care medicine and general pathology, this is nothing more than a special case of hemodynamic shock. Pro-inflammatory mediators, in relation to COVID-19, are referred to as a “cytokine storm” [11]. Moreover, in shock associated with polytrauma, infectious-septic or immunopathological factors, excessive systemic action of pro-inflammatory mediators is a very early link of the pathogenesis [30].

Similar processes also occur in shock-like states (in particular, those associated with hyperstimulation of immune-inflammatory mechanisms—hemophagocytic syndrome, hyperferritinemic syndrome and side effects of immunotherapy for oncological diseases, for example, chimeric antigen receptor of T-cells) [26,27,50]. At the same time, some studies of atherogenesis reported transformation of regulatory T-lymphocytes of the arterial wall from the initial protective phenotype (FoxP3+), which restrains autoimmune inflammatory processes, into the pathogenic one (RORyt, T-bet, Bcl-6), which contributes to the progression of the atherosclerotic plaque formation [51]. Such transformation is facilitated by high concentrations of pro-inflammatory autacoids, and COVID-19, obviously, can contribute to this process. Elevated serum levels of the C5a serine protease, which is important for the complement-mediated pro-inflammatory response, is a potential biomarker of disease severity in patients with COVID-19. The deposition of the terminal complement complex C5b-9 on endothelial cells promotes the release of thrombotic factors, triggering the production of pro-inflammatory cytokines [52]. Thrombus formation with recruitment and activation of neutrophils serves as a source of neutrophil extracellular traps (NETs). Like the serum level of circulating components of the complement system, serum levels of NETs positively correlate with the severity of both COVID-19 and atherosclerosis due to their cytotoxic effect on endothelial cells [53].

The clinical course of severe COVID-19 is characterized by an aberrant inflammatory response, in which the barrier function of inflammation is not observed:The systemic concentrations of pro-inflammatory mediators increase several times. Bioregulators begin to interfere with the regulation of systemic vital processes, conflicting with their central neuroendocrine regulation [50,54,55];Breathing, blood circulation and rheology are disturbed. The microcirculatory bed, even in organs without primary local lesions, acquires features of inflammatory foci (sticky endothelium, marginal standing of leukocytes, microthrombosis, erythrocyte sludge, disseminated intravascular coagulation);Stasis, increased vascular permeability and extravasation of plasma into the “third space”, decreased venous return to the heart, organ hypoperfusion and, ultimately, hemodynamic shock, i.e., hypoxic multiple organ failure [55];Self-sustaining process of a drastic increase in the production of cytokines initiated by IL-1;Systemic concentrations of pro-inflammatory autacoids (IL-1, IL-6, IL-10, IFNγ, G-CSF, MCP1, MIP -1α, PDGF, VEGF, ferritin, C-reactive protein, components of the complement, kinin, coagulation and fibrinolytic systems) in the blood increase dramatically [12,36,46].

The pathogenic role of hypercytokinemia, also regarding the cardiovascular system, has been repeatedly proven in COVID-19. Thus, the results of a retrospective, multicenter study confirmed that elevated systemic concentrations of pro-inflammatory bioregulators, including ferritin and IL-6, were associated with a more severe course of COVID-19 and multiple organ damage [56,57]. Another group of researchers confirmed that patients with severe COVID-19 have elevated serum levels of IL-2, IL-6, IL-10 and TNF-α [58]. The place of pro-inflammatory autacoids in COVID-19-associated acute cardiac injury is determined in Figure 1.

Hypercytokinemia is a well-known link in the pathogenesis of arterial hypertension, which in turn is a known risk factor for atherogenesis [59]. Cardiomyocytes death as a possible consequence of hypercytokinemia has already been discussed above. The degree of myocardial dysfunction in arterial hypertension also depends on the serum levels of pro-inflammatory cytokines [60], and therefore may increase in COVID-19 patients and survivors.

It is important that many of the pro-inflammatory mediators mentioned above are the key signals in atherogenesis and its thrombotic complications [7]. They contribute to the destabilization of atheromas, occlusive thrombus formation and vasoconstriction, which can lead to myocardial infarction in acute COVID-19 and the post-COVID period [3,17].

From the standpoint of general pathology, one should not forget that in case of hemodynamic shock of any etiology (for example, in multiple trauma), the hyperinflammatory phase is naturally replaced by the phase of immunosuppression and the production of anti-inflammatory signals. During this period, antimicrobial resistance declines and the risk of septic processes increases [61]. Atherogenesis, unlike COVID-19, is not a systemic process but is a local process affecting loca minimorum resistentia in the arteries against a systemic background of hyperlipidemia. Atheromas, however, are the inflammatory foci, because if they contain significant amounts of extracellular lipids and active macrophages, the plaques become destabilized; that means they become a source of vasospastic, thrombogenic and pro-inflammatory mediators. These mediators can act outside the focus itself, exactly what occurs in COVID-19, and provoke complications [7]. In atherogenesis, however, as in the course of shock and shock-like conditions, there is an anti-regulatory phase, when not only atherogenic, but also anti-atherogenic bioregulators increase in atherosclerotic lesions. The main anti-atherogenic cytokines are TGF-β and several interleukins (IL-5, IL-10, IL-13, IL-19, IL-27, IL-33, IL-35, IL-37), which enhance the activity of Treg cells and reduce the production of proatherogenic TNFα [62]. Whether they also serve as factors of sanogenesis in COVID-19 remains to be seen, but already there are some studies on this subject. Thus, IL-32 provides sanogenic effects during atherogenesis through differential regulation of macrophage polarization at its different stages, possibly depending on various stimuli that occur inside the plaque at different stages of its development. IL-32 suppresses the activity of CCL-2 and MMP, and its expression is increased in unstable plaques. It has been speculated that changes in its production may explain the tendency towards destabilization of atheromas observed during and after COVID-19 [45]. IL-34 may contribute to atherosclerosis, but its role in COVID-19 remains unclear. It is believed that IL-34 can be upregulated in tissues during their infiltration by leukocytes, especially mononuclear cells, during cytokine storm in patients with COVID-19 and contribute to the destabilization of atherosclerotic plaques [45]. Circulating IL-37 level has been reported to increase in patients infected with COVID-19. However, patients with higher serum levels of IL-37 had a shorter hospital stay, suggesting a sanogenic effect of this cytokine during COVID-19 infection. Interestingly, it also exhibits a stabilizing effect on atherosclerotic plaques [45].

## 8. Microvascular Dysfunction

It is suggested that vasculitis and thrombosis of the aortic vasa vasorum may represent key factors linking severe forms of SARS-CoV-2 infection with transformation of a stable plaque into an unstable lesion with thromboembolic consequences [63]. The impact of SARS-CoV-2 on large vessels such as the aorta is more complex than has been previously considered. We can assume that the virus targets not only on the intima of the artery, where it can cause inflammatory processes and endothelial dysfunction, but also the adventitia, where it triggers inflammation of the vasa vasorum (vasa vasoritis). This process can reduce the amount of blood, oxygen and nutrients supplying the media, and thereby promote atherogenesis [64].

Pro-inflammatory factors such as macrophages and granulocytes can infiltrate atheroma via intraplaque newly formed vessels originating from the vasa vasorum. These new vessels within plaques are often immature and therefore more permeable. These peculiarities of plaque neovascularization, in turn, predispose to the formation of intraplaque hemorrhage, which accelerates the progression of atherosclerotic lesions and creates the risk of thrombogenic complications. Intraplaque hemorrhage can enhance not only the inflammatory process but also the further development of neovascularization from the vasa vasorum due to the release of platelet growth factors, pro-inflammatory cytokines and other angiogenic stimuli, resulting in the pathogenetic vicious circle. Inflammatory cells are also an important source of matrix metalloproteinases (MMPs). The latter degrade the extracellular matrix to facilitate cell migration and recruitment. However, such degeneration weakens the fibrous elements of atheromas, which leads to the risk of their rupture [65].

The systemic action of inflammatory mediators against the background of blood flow centralization, as in severe COVID-19, favors coronary blood flow. However, if there are atherosclerotic plaques in the coronary bed, this is associated with the risk of their destabilization and rupture, paradoxically leading to myocardial ischemia and myocardial infarction. In addition, against the background of sepsis-induced hemodynamic shock, which results from a cytokine storm [55], the total reserves of coronary blood flow are reduced, and microvascular resistance indices increase in patients with acute coronary syndrome [52]. In atherosclerosis, as in COVID-19, extracellular vesicles can contribute to thrombosis or dissemination of the underlying process, depending on their composition and microenvironment [66].

Dozens of cases of a Kawasaki-like disease with arteritis, coronaritis and thrombosis have already been described during acute COVID-19 or in the post-COVID period. This is one of the typical autoimmune complications of COVID-19. However, the Kawasaki syndrome itself is a disease that de facto affects the vessels of the same caliber as atherosclerosis, for example, the coronary arteries, and is associated with the acceleration of atherogenesis. The same is probably true for COVID-19 [67]. In any case, lymphocytic perimyocarditis with coronary arteries vasculitis has been described in patients who died from COVID-19 [68,69].

## 9. Therapy Perspectives

Interestingly, pharmacological agents commonly used in the treatment of atherosclerosis, such as statins and acetylsalicylic acid, seem to reduce both the incidence of serious complications of COVID-19 and the death rate from this infection. The possible effects of chronic therapy with acetylsalicylic acid have not been fully studied. However, it has only a very limited anti-inflammatory effect at the low doses used in CVD. Therefore, patients with CVD should not stop taking these drugs for secondary prevention. In patients with COVID-19, statin therapy has been reported to be more likely to be complicated by side effects such as hepatic cytolysis or even severe rhabdomyolysis, and therefore it may be appropriate to temporarily suspend statin therapy during the acute infection [70]. Despite the understanding of the key role of endothelial dysfunction in the early stages of both diseases compared here, radical endothelial-related treatments have not been proposed for either COVID-19 or atherosclerosis [62]. There are also significant differences in the practice of using drugs in these two diseases. In severe COVID-19, when the adrenal glands are affected in a large percentage of cases by the virus, glucocorticoids (due to their anti-inflammatory, immunosuppressive and anti-shock antihypoxic effects) have an effective therapeutic effect, curbing the excessive systemic effect of inflammatory autacoids [55]. However, atherosclerosis is not treated with glucocorticoids, because their excess has a rather proatherogenic effect on lipid metabolism, and chronic stress and hypercorticism are considered risk factors for atherosclerosis [7,71]. Cardiotoxicity of some antiviral drugs used to treat COVID-19 should be also mentioned [72].

Despite the short experience of the COVID-19 pandemic, significant progress has been made in understanding the complex, mutually potentiating pathogenesis of cardiovascular injury in novel coronavirus infection. The identified main pathogenetic mechanisms represent therapeutic goals. Knowledge of the mechanisms of pathogenesis and their relationships allows them to be used as the main targets for protecting the cardiovascular system from the damaging effects of COVID-19. First of all, this refers to the need to stabilize the RAAS, prevent the penetration of the virus into cells, minimize the manifestations of endothelial dysfunction, normalize microcirculation and prevent thrombosis, combat hypoxia as a multicomponent process of mixed etiology and regulate immune and autoimmune processes. Currently, there are no clinical guidelines for cardio- and vasoprotection in the treatment of COVID-19. Additionally, in view of the relatively recent onset of the pandemic, it is not clear how the coronavirus infection will affect the population with atherosclerosis and its complications in the long term, although there is already reason to include COVID-19 in the list of infectious and immunopathological triggers of CVD.

The pathogenetic links discussed above are intermingled and reinforce each other (Figure 2).

Thus, the severe consequences of COVID-19 for the cardiovascular system can be explained by progressive multilevel equifinal damage, rather than dependence on alteration of a unique therapeutic target. Similarly, in atherogenesis, there is a whole continuum of risk factors [7], so it is impossible to stop atherogenesis by affecting one of them, even such a significant one as hyperlipidemia [64].

## 10. Conclusions

In any case, a group of experts from the European Society of Cardiology has recently recognized the long-term increased risk of cardiovascular disease after COVID-19, in particular due to the endotheliotropism of SARS-CoV-2 [69], but, as can be seen from the above, is supported by other mechanisms. Table 2 presents the various mechanisms by which COVID-19 contributes to atherogenesis and destabilization of atherosclerotic plaque.

Some practically significant aspects of the interaction between COVID-19 and the clinical manifestations of atherosclerosis in healthcare are not biological but are social in nature. Thus, a meta-analysis of 27 international studies showed that against the backdrop of a pandemic, there was a 40–50% decrease in the number of hospitalizations for acute coronary syndrome and an increase in the time “from door to device” for cardiac patients. The observed increase in pre-hospital mortality and out-of-hospital cardiac arrests indicates the negative impact of the pandemic on overall mortality rates from acute myocardial infarction, associated with a decrease in the availability of qualified inpatient cardiac care for somatic patients in the COVID-19 pandemic. In other words, one disease interfered with the control of another not only at the pathophysiological level but also at the medical and social levels [70].

This review aims to generalize the pathophysiology of COVID-19, an unexplored field that we have encountered recently, to the circulatory system. It reflects the relationship, in particular, with atherosclerosis, such a common and aggravating disease, while the respiratory system was previously the focus of treatment. In the future, we will have to live with this viral disease as seasonal. Thus, for public health, the literature sources of the most effective therapeutic prophylaxis are:(1)early appointment of an IL-6 inhibitor in case of damage to cardiomyocytes by hypercytokinemia;(2)complex treatment of any type of hypoxia (from respiratory to histological);(3)search for methods to eliminate endothelial dysfunction;(4)study of the effect of RAAS blockers on the prevention of thrombosis;(5)monitoring the level of circulating IL-32 and IL-37 as biological markers that exhibit sanogenic effects and stabilize atherosclerotic plaques;(6)normalization of microcirculation to prevent the transformation of a stable plaque into an unstable one.

## Figures and Tables

**Figure 1 diagnostics-13-00478-f001:**
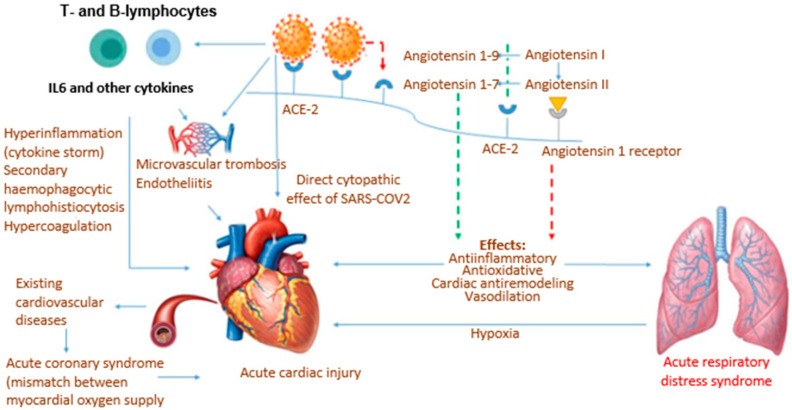
The relationship between potential mechanisms of acute cardiac injury in COVID-19 with the focus on pro-inflammatory autacoids.

**Figure 2 diagnostics-13-00478-f002:**
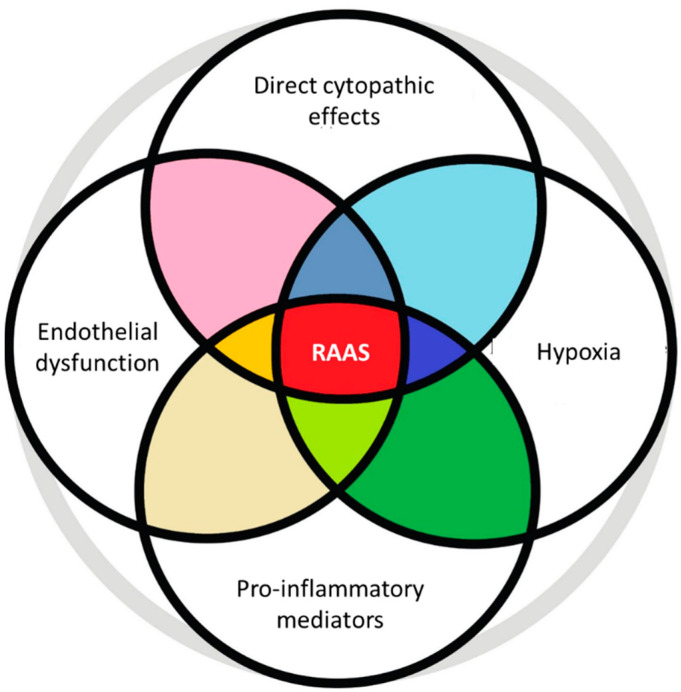
Mutual potentiation of the main pathophysiological mechanisms triggered by SARS-CoV-2 in case of damage to the cardiovascular system.

**Table 1 diagnostics-13-00478-t001:** Mechanisms of COVID-19-related cardiovascular injury.

Mechanisms	References
Direct cytopathic effect of SARS-CoV-2 on the cardiomyocytes and myocardial fibrosis as the consequence of COVID-19-induced myocarditis	[11,12,13]
Endothelial dysfunction and coagulopathy	[4,14,15]
A mismatch between myocardial oxygen supply and demand caused by cardiac overstrain as a compensatory response to hypoxia	[12]
Systemic inflammatory response, which can result in cytokine storm and hemodynamic shock	[7,16]
Renin–angiotensin–aldosterone system dysfunction	[2,17,18]

**Table 2 diagnostics-13-00478-t002:** Mechanisms of COVID-19’s contribution to atheroma formation and development.

Mechanisms	References
Reduction of the level of ACE2 which prevents the degradation of pro-atherosclerotic angiotensin II and generation of antiatherosclerotic angiotensin 1–7.	[45]
Direct cytopathic effect of the virus on endothelial cells	[69]
Protease activation, causing a transition from a stable to a pathological atherosclerotic injury and the degradation of the plaque protective fibrous cap.	[9,42]
Leucocyte recruitment and adhesion to the vascular wall	[12]
Expression of pro-inflammatory chemokines and cytokines (CXCL1, CXCL2, CXCL3, CXCL8, IL-1β) in the vascular wall resulted from the activation of MyD88-dependent pathways	[1,34]
VEGLX damage	[36]
Mitochondrial dysfunction leading to the increased production of reactive oxygen species	[33]
Endothelial damage caused by neutrophil extracellular traps (NETs)	[53]
Endothelial damage caused by the components of the activated complement system	[52]
Vasa vasorum vasculitis and/or thrombosis	[63]

## Data Availability

Not applicable.

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
