# Peer review of "Atherosclerosis, Cardiovascular Disorders and COVID-19: Comorbid Pathogenesis"

_diagnostics, 2023, doi:10.3390/diagnostics13030478_

Round 1
Reviewer 1 Report
The current manuscript titled: "Aterosclerosis, Cardiovascular Disorders And Covid-19: Comorbid Pathogenesis" represents an important analysis of evolving field of Vascular Medicine, Internal Medicine and Infectious Diseases.
The title reflects the manuscript content and helps the reader navigate the article essence.
In my opinion, these are the adjustments which should be made to increase the value of your manuscript:
1. First, I would like to emphasize that typos made in line 17 (…ACE2 принимая во внимание его влияние на, 17 other mechanisms…) are inadmissible for articles submitted to ISI-level Journals. I hope the Authors will take this into account. Please change this sentence.
2. Change please “Covid-19” to “COVID-19”.
3. In abstract, please, add abbreviation for “SARS-CoV-2”, “ACE2”, “LDL”.
4. In the main manuscript text, starting with the Introduction section, please add abbreviations to all shortened words (e.g., COVID-19, SARS-CoV-2, ECG, etc.).
5. In Table 1, change please “SARS-COV2” to “SARS-CoV-2”.
6. What methodology was used to select the bibliography for Table 1? Please specify.
7. Change please “renin-angiotensin-aldosterone systeme” to “renin-angiotensin-aldosterone system”.
8. Please, add more detailed information about the pathophysiological processes linking renin-angiotensin-aldosterone system and COVID-19.
9. In the Conclusions section, highlight the practical utility and significance of this review.
10. It is recommended that the Therapy Perspectives subsection be separated from the Conclusions section and labeled separately so that the Conclusions section comes last.
11. The manuscript contains some punctuation errors and typos, please revise the text (lines 26, 44, 52, 122, 160, etc.).
12. Adapt the references according to the Journal requirements.
Author Response
Response to Reviewer 1 Comments
Thank you for your letter and for the opportunity to revise our paper “Aterosclerosis, Cardiovascular Disorders And Covid-19COVID-19: Comorbid Pathogenesis”. The suggestions offered by both reviewers have been immensely helpful, and we also appreciate your insightful comments.
Point 1
First, I would like to emphasize that typos made in line 17 (…ACE2 принимая во внимание его влияние на, 17 other mechanisms…) are inadmissible for articles submitted to ISI-level Journals. I hope the Authors will take this into account. Please change this sentence.
Change please “Covid-19” to “COVID-19”.
In abstract, please, add abbreviation for “SARS-CoV-2”, “ACE2”, “LDL”.
In the main manuscript text, starting with the Introduction section, please add abbreviations to all shortened words (e.g., COVID-19, SARS-CoV-2, ECG, etc.).
In Table 1, change please “SARS-COV2” to “SARS-CoV-2”.
Change please “renin-angiotensin-aldosterone systeme” to “renin-angiotensin-aldosterone system”.
The manuscript contains some punctuation errors and typos, please revise the text (lines 26, 44, 52, 122, 160, etc.).
It is recommended that the Therapy Perspectives subsection be separated from the Conclusions section and labeled separately so that the Conclusions section comes last.
Adapt the references according to the Journal requirements
Response 1
We agree with the Reviewer. The indicated changes have been implemented.
Point 2
What methodology was used to select the bibliography for Table 1? Please specify.
Response 2
Thank You for this question. We searched PubMed for the open-access papers addressing pathophysiology of the COVID-19-related cardiovascular injury, and also included in the bibliography some monographs of our colleagues, who has been working on this issue since the onset of the COVID-19 pandemic.
Point 3
Please, add more detailed information about the pathophysiological processes linking renin-angiotensin-aldosterone system and COVID-19.
Response 3
We are grateful for Your interest in this subject and added the following information:
“It is known that SARS-CoV-2 uses ACE2, a negative regulator of the renin-angiotensin-aldosterone system (RAAS) as a receptor for entry into cells, including almost all cell types in heart and blood vessel wall. ACE-2 is a type I transmembrane protein that is expressed in the lungs (high level of expression on the surface of type II alveolar cells). The physiological role of ACE-2 is primarily associated with the breakdown of angiotensin I (Ang-II) to the inactive Ang-(1-9) peptide, which is further converted into Ang-(1-7), using ACE or other peptidases, which binds to Mas receptors. Ang-(1-7) provides vaso- and cardioprotection, antiproliferative, anti-inflammatory and natriuretic effects, and also has protective effects against heart failure, thrombosis, myocardial hypertrophy, fibrosis, arrhythmia, atherogenesis.”
Point 4
In the Conclusions section, highlight the practical utility and significance of this review.
Response 4
We agree with the Reviewer and added the following text.
“This review aims to generalize the pathophysiology of COVID-19, an unexplored field that we have encountered recently, to the circulatory system. It reflects the relationship in particular with atherosclerosis, such a common and aggravating disease, while the respiratory system was previously the focus of treatment. In the future, we will have to live as seasonal with this viral disease. Thus, for public health, a literary summary of the impact on various systems and the safest and most effective therapeutic regimens is needed”
Reviewer 2 Report
Reviewer comments and suggestions
The authors in this manuscript suggested the possible way of relationship between atherosclerosis and coronavirus disease 19 (COVID-19) A bidirectional influence of these conditions is described: the presence of cardiovascular diseases affects different individuals susceptibility to viral infection. The study also discussed that SARS CoV-2 could have a negative effect on the endothelium and cardiomyocytes, causing blood clotting, secretion of pro-inflammatory cytokines, and thus exacerbating the development of atherosclerosis.
They also describe the pathogenesis of the virus on various links which are required for atherogenesis such as the generation of oxidized forms of LDL, launch of a cytokine storm, damage to the endothelial glycocalyx, and mitochondrial injury. Additionally, the authors discussed the severity of SARS-CoV-2 on large vessels such as the aorta which is more and further adding the mutual influence of the medicines used in the treatment of atherosclerosis and acute COVID-19 was also explained in the manuscript.
Overall, the manuscript was well written. However, a few concerns/comments needed to be explained/modified.
- Line number 17, there were typo errors please check the line
- Line 32, please mention an appropriate reference
- Lines 54-55 need an appropriate reference for the line
- Line 63 the authors have to explain how in this sentence
- Line 81, the author suggested studies but did not mention the references there and too many sentences create confusion in the reading of your paper, idea should be short not long sentences (lines 82-86)
- Line 120-122 what would be the reason presented by the authors
- Appropriate references are needed for lines 123-128
- Mistake in the section numbers please check 3 it (3. Endothelial dysfunction in COVID-19 and atherosclerosis: common links) should be 4 that successively follows other sections
- Line 227-229 is these lines were important to add here, these links were already known in various models
- Section 6 or the comments can be applied to other sections as well ‘please add figures rather than explaining all, it will help to understand the readers, at least two figures are needed in the manuscript’
- Comments for conclusion paragraph: I am not convinced about the section, needs to revise and the figure should be placed somewhere else.
12. All references need to be modified, based on the MDPI guidelines
Author Response
Thank you for your letter and for the opportunity to revise our paper “Aterosclerosis, Cardiovascular Disorders And Covid-19COVID-19: Comorbid Pathogenesis”. The suggestions offered by both reviewers have been immensely helpful, and we also appreciate your insightful comments.
Overall, the manuscript was well written. However, a few concerns/comments needed to be explained/modified.
Point 1 Line number 17, there were typo errors please check the line
Mistake in the section numbers please check 3 it (3. Endothelial dysfunction in COVID-19 and atherosclerosis: common links) should be 4 that successively follows other sections
All references need to be modified, based on the MDPI guidelines
Response 1 We agree with the Reviewer. The indicated changes have been implemented.
Point 2 Line 32, please mention an appropriate reference
Lines 54-55 need an appropriate reference for the line
Appropriate references are needed for lines 123-128
Response 2 Thank You for Your interest in this subject. The appropriate references have been incorporated.
Point 3 Line 63 the authors have to explain how in this sentence
Response 3 We agree with the Reviewer and changed this sentence to:
“Therefore, a pro-inflammatory and thrombophilic state is an integral feature of atherosclerosis, potentially increasing vulnerability to severe COVID-19 because the underlying endothelial dysfunction might represent the ideal deregulated immunological setting in which SARS-CoV-2 triggers a ‘‘cytokine storm’’”.
Point 4 Line 81, the author suggested studies but did not mention the references there and too many sentences create confusion in the reading of your paper, idea should be short not long sentences (lines 82-86)
Response 4 We agree with the Reviewer, changed the sentence and added references:
“Studies [1,9,10] have shown that infection with SARS-CoV-2 can accelerate the course of both initial and advanced stages of atherogenesis and cause its clinical manifestation by different mechanisms. These mechanisms include activation of coagulation and thrombosis, secretion of pro-inflammatory cytokines and chemokines by endothelial cells, rise in the level of fibrinogen, antithrombin and D-dimer in the blood and, ultimately, triggering regional disseminated intravascular coagulation [10].”
Point 5 Line 120-122 what would be the reason presented by the authors
Response 5 Thank You for Your interest. We provided an additional sentence and described potential underlying mechanisms.
Point 6 Line 227-229 is these lines were important to add here, these links were already known in various models
Response 6 We consider these lines to be important for the coherence, as we address the role of cholesterol in COVID-19 further in the text.
Point 7 Section 6 or the comments can be applied to other sections as well ‘please add figures rather than explaining all, it will help to understand the readers, at least two figures are needed in the manuscript’
Response 7 We agree with the Reviewer, cut the section and added the figure which made Section 6 clearer for the reader
Point 8 Comments for conclusion paragraph: I am not convinced about the section, needs to revise and the figure should be placed somewhere else.
Response 8 We agree with the Reviewer. The Conclusion section has been revised and changed, and the Figure has been placed in another section.
Round 2
Reviewer 1 Report
The article lacks specific conclusions. The Authors repeatedly talk about goals and theory. It is recommended to edit the conclusions and emphasize the practical significance of this review.
Author Response
This review aims to generalize the pathophysiology of COVID-19, an unexplored field that we have encountered recently, to the circulatory system. It reflects the relationship in particular with atherosclerosis, such a common and aggravating disease, while the respiratory system was previously the focus of treatment. In the future, we will have to live as seasonal with this viral disease. Thus, for public health, the literature sources of the most effective therapeutic prophylaxis are:
- early appointment of an IL-6 inhibitor in case of damage to cardiomyocytes by hypercytokinemia;
- complex treatment of any types hypoxia (from respiratory to histological);
- search for methods to eliminate endothelial dysfunction;
- study of the effect of RAAS blockers on the prevention of thrombosis;
- monitoring the level of circulating IL-32 and IL-37 as biological markers that exhibit sanogenic effects and stabilize atherosclerotic plaques;
- normalization of microcirculation to prevent the transformation of a stable plaque into an unstable.
Round 3
Reviewer 1 Report
I recommend this article for publication.